# Diagnostic Accuracy of Urine Lipoarabinomannan Testing in Early Morning Urine versus Spot Urine for Diagnosis of Tuberculosis among People with HIV

Stephanie Bjerrum,[a,b] Johanna Åhsberg,[a,b] Rita Szekely,[c] Japheth Opintan,[d] Margaret Lartey,[e] Maunank Shah,[f] Satoshi Mitarai,[g] Tobias Broger,[c,h] Morten Ruhwald,[c] Isik Somuncu Johansen[a,b]

[a]University of Southern Denmark, Research Unit for Infectious Diseases, Odense, Denmark

[b]Odense University Hospital, Department of Infectious Diseases, Mycobacterial Centre for Research Southern Denmark MyCRESD, Odense, Denmark

[c]FIND, Geneva, Switzerland

[d]Department of Medical Microbiology, University of Ghana Medical School, Accra, Ghana

[e]Department of Medicine & Therapeutics, University of Ghana Medical School, Accra, Ghana

[f]Department of Medicine, Division of Infectious Diseases, Johns Hopkins University, John Hopkins University School of Medicine, Baltimore, Maryland, USA

[g]Department of Mycobacterium Reference and Research, Research Institute of Tuberculosis, Japan Anti-Tuberculosis Association, Tokyo, Japan

[h]Division of Tropical Medicine, Center for Infectious Diseases, Heidelberg University Hospital, Heidelberg, Germany

**ABSTRACT** The Fujifilm SILVAMP TB LAM (FujiLAM) assay offers improved sensitivity compared to Determine TB LAM Ag (AlereLAM) for detecting tuberculosis (TB) among people with HIV. Here, we examined the diagnostic value of FujiLAM testing on early morning urine versus spot urine and the added value of a two-sample strategy. We assessed the diagnostic accuracy of FujiLAM on cryopreserved urine samples collected and stored as part of a prospective cohort of adults with HIV presenting for antiretroviral treatment in Ghana. We compared FujiLAM sensitivity and specificity in spontaneously voided urine samples collected at inclusion (spot urine) versus in the first voided early morning urine (morning urine) and for a one (spot urine) versus two samples (spot and morning urine) strategy. Diagnostic accuracy was determined against both microbiological (using sputum culture and Xpert MTB/RIF testing of sputum and urine to confirm TB) and composite reference standards (including microbiologically confirmed and probable TB cases). Paired urine samples of spot and morning urine were available for 389 patients. Patients had a median CD4 cell count of 176 cells/$\mu$L (interquartile range [IQR], 52 to 361). Forty-three (11.0%) had confirmed TB, and 19 (4.9%) had probable TB. Overall agreement for spot versus morning urine test results was 94.6% (kappa, 0.81). Compared to a microbiological reference standard, the FujiLAM sensitivity (95% confidence interval [CI]) was 67.4% (51.5 to 80.9) for spot and 69.8% (53.9 to 82.8) for morning urine, an absolute difference (95% CI) of 2.4% (−10.2 to 14.8). Specificity was 90.2% (86.5 to 93.1) versus 89.0% (85.2 to 92.1) for spot and morning urine, respectively, a difference of 1.2% (−3.7 to 1.4). A two-sample strategy increased FujiLAM sensitivity from 67.4% (51.5 to 80.9) to 74.4% (58.8 to 86.5), a difference of 7.0% (−3.0 to 16.9), while specificity decreased from 90.2% (86.5 to 93.1) to 87.3% (83.3 to 90.6), a difference of −2.9% (−4.9 to −0.8). This study indicates that FujiLAM testing performs equivalently on spot and early morning urine samples. Sensitivity could be increased with a two-sample strategy but at the risk of lower specificity. These data can inform future guidelines and clinical practice.

**IMPORTANCE** This study indicates that FujiLAM testing performs equivalently on spot and early morning urine samples for detecting tuberculosis among people with HIV. Sensitivity could be increased with a two-sample strategy but at the risk of lower specificity. These data can inform future guidelines and clinical practice around FujiLAM.

Address correspondence to Stephanie Bjerrum, sbjerrum@health.sdu.dk.

The authors declare a conflict of interest. No disclosure from S.B., I.S.J., J.A., J.O., M.L. For T.B., M.R., R.S., they were affiliated with FIND during the conduct of the study. T.B. reports patents in the field of TB detection and is a shareholder of Avelo Inc. S.M. reports collaboration with FujiFilm for the development of SILVAMP TB LAM and that the study budget received approximately 20,000 USD in the last two years for conduct of FujiLAM tests at RIT/JATA laboratory under the contract with FIND. The funders of the study had no role in study design, data collection, data analysis, data interpretation, or writing of the manuscript. FIND is a not-for-profit NGO that collaborates in partnerships to develop, evaluate and implement new diagnostics for LMIC. FIND has product evaluation agreements with FujiFilm and several other private sector companies that design diagnostics and related products for treatment of tuberculosis and other diseases. These agreements strictly define FIND's independence and neutrality vis-à-vis the companies whose products get evaluated and describe roles and responsibilities.

**KEYWORDS** diagnostic accuracy, urine, LAM, tuberculosis, HIV, accuracy, lipoarabinomannan

Diagnosing tuberculosis (TB) among people with HIV (PWH) remains a major challenge, and TB is often left undiagnosed (1). PWH may have paucibacillary disease, be unable to produce sputum samples, or present with extrapulmonary TB (2). Molecular rapid diagnostics, such as the Xpert MTB/RIF assay (Cepheid, Sunnyvale, CA, US) or the more recent Xpert MTB/RIF Ultra assay, are recommended by World Health Organization (WHO) as initial TB diagnostic tests (3 to 5), but implementation of these assays has been challenged by costs, instrumentation requiring stable electricity, and maintenance, which are hurdles for point of care (6, 7).

An alternative strategy for TB diagnosis in PWH is the detection of the lipoarabinomannan (LAM) antigen in urine using immunoassays (8). Since 2015, WHO has recommended use of the point-of-care test Determine TB LAM Ag (AlereLAM; Abbott, Palatine, IL, USA) for TB diagnosis among subgroups of PWH, and expanded guidance was released in 2019 (9, 10). So far, AlereLAM is the only marketed LAM test, with a pooled sensitivity of 42% among patients, with symptoms increasing to 54% for those with CD4 counts equal to or below 100 cells/$\mu$L against a microbiological reference standard. However, sensitivity among individuals with CD4 counts above 100 cells/$\mu$L was only 17% (11). Despite suboptimal sensitivity, several studies have found an impact of AlereLAM on clinically important outcomes (12–14). The new rapid urine LAM assay, Fujifilm SILVAMP TB LAM (FujiLAM; Fujifilm, Tokyo, Japan), is developed to detect urinary LAM at lower concentrations than AlereLAM by using improved detection antibodies and a silver amplification step to increase the visibility of test results (15). We and others found an increased absolute sensitivity of FujiLAM for TB diagnosis among PWH by ~30% compared to AlereLAM across CD4 strata for a microbiological reference standard (16, 17). Sensitivity of FujiLAM in some studies reaches 70% to 85% and has been most studied among adult PWH (16, 18).

While there is evidence that accuracy of FujiLAM is nearly equivalent in fresh versus frozen samples (19), the number and type of urine samples required for optimal FujiLAM testing has not been assessed. To inform guidelines and policy development around the use of FujiLAM, it will be important to assess if the diagnostic accuracy of FujiLAM is affected by the timing of sample collection and if accuracy can be enhanced by conducting a second FujiLAM test on an additional sample. With this study, we aim to determine the sensitivity and specificity of FujiLAM testing of spot urine versus early morning urine and assess the diagnostic value of a two-samples strategy (one spot and one morning urine sample) to detect TB in a cohort of PWH commencing antiretroviral treatment (ART) in Ghana.

## RESULTS

From the DETECT-TB study cohort, 532 of 568 participants had data available for classification of TB status and retrospective FujiLAM testing, but 143 were excluded, as they only had a single urine sample available for testing. The remaining 389 participants had paired spot and early morning urine samples (354 outpatients and 35 inpatients) (Fig. 1). The median time interval (in days) between spot and morning urine was 3 days (interquartile range [IQR], 2 to 5). Participants were characterized by female sex (257/389; 66.0%) with 18/257 (7.0%) being pregnant, a median age of 38 years (IQR, 31 to 45), and a median CD4 cell count of 176 cells/$\mu$L (IQR, 52 to 361) (Supplementary Table S1). The majority of participants 329/389 (84.6%) were WHO symptom screen positive with the presence of any of fever, cough, weight loss, or night sweats. Based on a combination of microbiological and clinical data, 43 (11.0%) participants were classified as confirmed TB, 19 (4.9%) as probable TB, and 327 (84.1%) as no TB (Fig. 1). Of participants with confirmed TB, 28/43 (65.1%) were identified by Xpert (sputum or urine) and 37/43 (86.0%) by sputum culture. Participants with probable TB were all started on empirical TB treatment, and three had a positive sputum smear for acid-fast

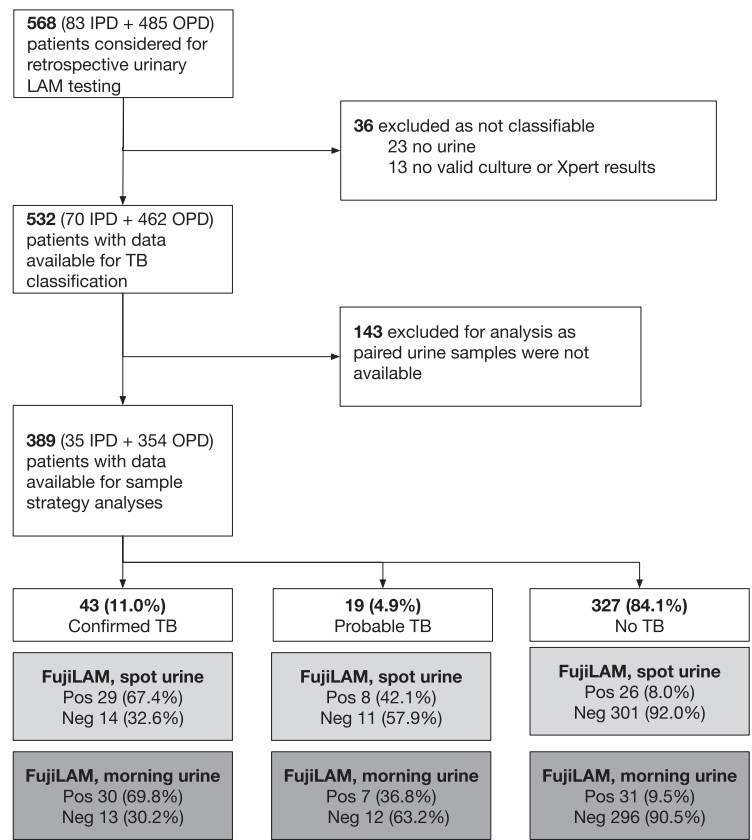

**FIG 1** Flow diagram of study population of TB diagnostic classification and FujiLAM test results. IPD, inpatients; LAM, lipoarabinomannan; OPD, outpatients; TB, tuberculosis.

bacilli during follow-up. No additional positive TB microbiological results (Xpert, culture, microscopy) were recorded during follow-up. Of participants classified as no TB, 52 (13.4%) were either lost to follow-up or had died before 2 months from enrollment and were excluded in sensitivity analysis. All-cause mortality was 5.7% (22 died) at 2 months and 9.2% (36 died) at 6 months.

**FujiLAM accuracy in spot versus early morning urine.** Among 43 participants with confirmed TB, 29 were identified by FujiLAM testing of the spot urine sample (sensitivity 67.4%; 95% confidence interval [CI], 51.5 to 80.9), and 30 were identified by testing the morning urine sample (sensitivity 69.8%; 95% CI, 53.9 to 82.8), an absolute difference of 2.4% (95% CI, −10.2 to 14.8); see Table 1. Specificity was 90.2% (95% CI, 86.5 to 93.1) versus 89.0% (95% CI, 85.2 to 92.1) for spot and morning urine, respectively, a difference of −1.2% (95% CI, −3.7 to 1.4). While 27 TB cases were positive in both samples, three TB cases were FujiLAM positive only in morning urine, and two cases only positive in spot urine.

When using a composite reference standard, sensitivity and specificity did not change substantially when comparing spot and morning urine sample test results (Table 1).

**FujiLAM accuracy in one- versus two-sample strategy.** In a two-sample strategy testing both spot and early morning urine compared to testing only spot urine, sensitivity increased from 67.4% (95% CI, 51.5 to 80.9) to 74.4% (95% CI, 58.8 to 86.5) (difference of 7.0%; 95% CI, −3.0 to 16.9) against a microbiological reference standard with an additional five TB cases identified (Table 1). Specificity for a two-sample strategy, however, decreased from 90.2% (95% CI, 86.5 to 93.1) to 87.3% (95% CI, 83.3 to 90.6), a difference of −2.9, (95% CI, −4.9 to −0.8) against a microbiological reference standard. Using a composite reference standard, testing two samples compared to one sample increased

**TABLE 1** Sensitivity and specificity of FujiLAM in spot and morning urine samples against the MRS and CRS[a]

| MRS or CRS | Test strategy | No. of samples | No. of TP results | No. of FP results | No. of FN results | No. of TN results | Sensitivity (% [95% CI]) | Specificity (% [95% CI]) | TB% | PPV (95% CI) | NPV (% [95% CI]) | △Sn (% [95% CI])[e] | △Sp (% [95% CI])[e] |
|---|---|---|---|---|---|---|---|---|---|---|---|---|---|
| MRS[b] | FujiLAM, spot | 389 | 29 | 34 | 14 | 312 | 67.4 (51.5 to 80.9) | 90.2 (86.5 to 93.1) | 11.1 | 46.0 (33.4 to 59.1) | 95.7 (92.9 to 97.6) | | |
| | FujiLAM, morning | 389 | 30 | 38 | 13 | 308 | 69.8 (53.9 to 82.8) | 89.0 (85.2 to 92.1) | 11.1 | 44.1 (32.1 to 56.7) | 96.0 (93.2 to 97.8) | 2.4 (−10.2 to 14.8) | −1.2 (−3.7 to 1.4) |
| | FujiLAM, two-sample[d] (any test positive) | 389 | 32 | 44 | 11 | 302 | 74.4 (58.8 to 86.5) | 87.3 (83.3 to 90.6) | 11.1 | 42.1 (30.9 to 54.0) | 96.5 (93.8 to 98.2) | 7.0 (−3.0 to 16.9) | −2.9 (−4.9 to −0.8) |
| CRS[c] | FujiLAM, spot | 389 | 37 | 26 | 25 | 301 | 59.7 (46.4 to 71.9) | 92.0 (88.6 to 94.7) | 15.9 | 58.7 (45.6 to 71.0) | 92.3 (88.9 to 95.0) | | |
| | FujiLAM, morning | 389 | 37 | 31 | 25 | 296 | 59.7 (46.4 to 71.9) | 90.5 (86.8 to 93.5) | 15.9 | 54.4 (41.9 to 66.5) | 92.2 (88.7 to 94.9) | 0.0 (−9.4 to 9.4) | −1.5 (−4.2 to 1.1) |
| | FujiLAM, two-sample[d] (any test positive) | 389 | 40 | 36 | 22 | 291 | 64.5 (51.3 to 76.3) | 89.0 (85.1 to 92.2) | 15.9 | 52.6 (40.8 to 64.2) | 93.0 (89.6 to 95.5) | 4.8 (−2.1 to 11.8) | −3.0 (−5.2 to −0.9) |

[a]CI, confidence interval; CRS, composite reference standard; FN, false negative; FP, false positive; FujiLAM, Fujifilm SILVAMP TB LAM assay; LAM, lipoarabinomannan; MRS, microbiological reference standard; NPV, negative predictive value; PPV, positive predictive value; Sn, sensitivity; Sp, specificity; TB%, tuberculosis prevalence; TN, true negative; TP, true positive.

[b]Using MRS, confirmed tuberculosis cases were considered reference standard positive. No tuberculosis and probable tuberculosis cases were considered reference standard negative.

[c]Using CRS, confirmed tuberculosis and probable tuberculosis cases were considered reference standard positive. No tuberculosis cases were considered reference standard negative.

[d]Spot urine was considered the reference for evaluation of two-sample strategy.

[e]△Sn and △Sn reported with the exact binomial 95% CI for sensitivity and specificity differences using McNemar's paired test of proportion.

**TABLE 2** Agreement of FujiLAM test results for spot versus morning urine[a]

| FujiLAM result | Morning negative | Morning positive | Overall percentage agreement (Kappa value, SE) |
|---|---|---|---|
| FujiLAM, all participants (n = 389) | | | |
| Spot negative | 313 | 13 | 94.6 (0.81, 0.05) |
| Spot positive | 8 | 55 | |
| | | | |
| FujiLAM, MRS positive[b](n = 43) | | | |
| Spot negative | 11 | 3 | 88.4 (0.73, 0.15) |
| Spot positive | 2 | 27 | |
| | | | |
| FujiLAM, CRS positive[c] (n = 62) | | | |
| Spot negative | 22 | 3 | 90.3 (0.80, 0.13) |
| Spot positive | 3 | 34 | |
| | | | |
| FujiLAM, not TB (n = 327) | | | |
| Spot negative | 291 | 10 | 95.4 (0.71, 0.06) |
| Spot positive | 5 | 21 | |

[a]CRS, composite reference standard; MRS, microbiological reference standard; SE, standard error; TB, tuberculosis.
[b]MRS positive, confirmed tuberculosis were considered reference standard positive.
[c]CRS positive, confirmed tuberculosis and probable tuberculosis were considered reference standard positive.

sensitivity from 59.7% to 64.5% (difference of 4.8%; 95% CI, −2.1 to 11.8,) while specificity dropped from 92.0% to 89.0% (difference of −3.0%; 95% CI, −5.2 to −0.9).

**FujiLAM accuracy stratified by patient characteristics.** The performance of FujiLAM testing strategies for subpopulations stratified by CD4 cell count, patient status (inpatient and outpatients), advanced HIV disease, and among those who died within 6 months is shown in Supplementary Table S3. The numbers are generally small, and testing morning urine compared to spot urine did not change sensitivity or specificity significantly across subpopulations except for women, where we found that specificity dropped significantly from 93.6% in spot urine to 90.3% in morning urine (difference of −3.4; 95% CI, −6.7 to −0.1) against a microbiological reference standard, while sensitivity remained similar. For men, sensitivity and specificity remained unchanged for spot and morning urine. For a two-sample test strategy compared to testing one sample, the trend was toward enhanced sensitivity for the more immunocompromised individuals, while no increased sensitivity was observed for those with CD4 cell count of >200 cells/$\mu$L. The additional five TB cases identified by a two-sample strategy were women and mainly identified among the subpopulations with CD4 counts ≤200 cells/$\mu$L. A two-sample strategy identified 90.0% of all TB cases among those who died before 6 months and all TB cases among inpatients. Specificity was lower for a two-sample strategy than a one-sample strategy for the subpopulations with low CD4 cell count and was 66.4%, 98.3%, and 97.5% for patients having CD4 counts below 100 (Supplementary Table S3).

In sensitivity analyses excluding 52 participants with no TB who died or were lost to follow-up before 2 months did not change overall accuracy results for early morning urine nor for a two-sample strategy (Supplementary Table S4).

**FujiLAM agreement in spot and early morning urine.** Overall, FujiLAM was positive in 63 (16.2%) of spot urine samples and 68 (17.5%) of early morning urine samples, with discordant results in 21 participants. Overall agreement of results from spot versus morning urine was high at 94.6% (kappa, 0.81; standard error [SE], 0.05), and agreement is provided by TB classification in Table 2.

Characteristics of participants stratified by FujiLAM results are shown in Supplementary Table S1. While participants with FujiLAM-positive results in both samples were related to confirmed TB status and a positive urine Xpert MTB, discordant FujiLAM results with one positive and one negative result were associated with female sex. Of note, all participants that had negative spot urine and positive morning urine FujiLAM results (n = 13) were women (3 confirmed TB and 10 no TB cases), while no other host characteristics like

pregnancy, proteinuria, or creatinine were associated with discordant results. The individual characteristics for the FujiLAM-discordant patients are provided in Supplementary Table S2.

## DISCUSSION

Based on cryopreserved samples, this study is the first to compare and demonstrate near-equivalent sensitivity and high agreement of FujiLAM testing of spontaneously voided urine sample compared to early morning sample for diagnosis of TB among a cohort of PWH. Testing two samples compared to one sample indicated a trend toward increased FujiLAM sensitivity, mainly among severely immunocompromised subpopulations, but at the cost of lower specificity.

Diagnostic accuracy studies of FujiLAM have repeatedly demonstrated superior sensitivity to AlereLAM in comparative accuracy studies conducted among PWH (17) and, more recently, also among adults without HIV (20, 21) and pediatric populations (22). AlereLAM is, to date, the only commercially available WHO endorsed point-of-care biomarker assay (8). FujiLAM is expected to become available commercially, and several prospective studies of FujiLAM are ongoing (ClinicalTrials.gov identifiers NCT04089423 and NCT04545164). To inform guidelines and policy development, it is of interest to identify strategies to maximize utility of FujiLAM testing for TB diagnosis while minimizing financial and human resource requirements for testing. We found little or no added benefit of testing morning urine compared to spot urine. The overall categorical agreement was high for spot and early morning urine and moderate when restricting analysis to those with confirmed TB. Although not statistically significant, we did see a trend toward a higher diagnostic yield of morning specimens than spot specimens for predefined subpopulations that are more severely sick as determined by inpatient status, low CD4 cell count, advanced HIV, or early mortality. This is the subpopulation that may also benefit from a two-sample strategy.

Overall, testing of one urine sample identified 67.4% of all microbiologically confirmed TB cases, and testing two samples identified 74.4% of confirmed TB cases and all TB cases among inpatients. A one-sample strategy and independence of timing for urine collection for LAM testing may simplify implementation of FujiLAM testing. One feasibility study of AlereLAM found that the short turnaround time permitted same-day initiation of TB treatment (23), and it needs to be seen if that can also be achieved with FujiLAM, which has a longer time to result. A requirement for patients to provide a morning sample or await two samples could prolong the diagnostic pathway with a risk of losing patients to follow-up. Although our results must be interpreted with caution due to limited sample size, in particular for subgroup analysis, our data do not warrant the programmatic challenge of an additional early morning urine sample in general. It may, however, be prudent to offer a second FujiLAM test on early morning urine for the most immunocompromised or hospitalized patients, who would also derive the biggest benefit and for whom the test is most sensitive.

The specificity of FujiLAM was generally not affected by whether testing was done on spot or morning urine samples but decreased significantly with a two-sample strategy. Test specificity was particularly lower for a two-sample testing strategy for subpopulations with low CD4 cell count. This may reflect the suboptimal reference standard and concerns of misclassification bias as has been discussed previously (16). A more comprehensive evaluation for extrapulmonary TB could have revealed more TB cases among FujiLAM-positive patients, thereby increasing diagnostic specificity with less heterogeneity across subpopulations. Whether there are other biological, laboratory, or clinical factors that may lead to falsely positive FujiLAM results remains an area of interest for future studies. Surprisingly, all 13 cases with a FujiLAM-positive morning but negative spot urine results were women, and all additional five TB cases identified with a two-sample strategy were women. For AlereLAM, female sex was also found to be associated with a discordant AlereLAM test result. We did not expect this finding and have not seen similar reports in previous FujiLAM or AlereLAM studies. One study of TB

LAM enzyme-linked immunosorbent assay (ELISA) reported higher sensitivity among females than males (67% versus 38%, $P$ = 0.023) and lower specificity (83.7% versus 93.9%) (24). The same study group also suggested that dust, soil, and stool led to false-positive results for the TB LAM ELISA (25). Previous Alere or FujiLAM studies have not reported accuracy by gender. Some assessed but did not find gender to be associated with AlereLAM positivity in spot urine samples (26–28). To understand this finding, it may be relevant to explore further both clinical, biological, and preanalytical factors related to collection and storage of samples, including an assessment of sex as a variable to explain heterogeneity in FujiLAM accuracy. This could be done by use of existing data set and attention on gender in future LAM studies.

While our study did not intend to compare AlereLAM and FujiLAM results for multiple test strategies, we observed a higher agreement of FujiLAM results for spot and morning urine than for AlereLAM. Few studies examined the value of early morning urine samples for the AlereLAM assay. While Gina et al. found that AlereLAM sensitivity increased from 12% (5/41) to 39% (16/41) when using early morning urine compared to a random urine sample (29), we report an increase in AlereLAM sensitivity from 44.2% (19/43) in spot urine to 53.5% (23/43) in early morning urine. The larger gain in sensitivity seen in the study by Gina et al. may relate to a sicker study population of hospitalized patients with HIV initiating anti-TB treatment (median CD4 count of 88 cells/$\mu$L) versus our cohort of mainly outpatients with HIV assessed for TB (median CD4 count of 176 cells/$\mu$L). The study by Gina et al. did not include gender as a variable in their analyses of predictors for increased AlereLAM sensitivity in early morning urine and did not assess specificity.

Quantitative studies of urine LAM have previously shown that LAM test results correlate with degree of bacillary burden (30–32), MGIT time to detection, Xpert MTB/RIF semiquantitative results, and smear status (33). Moreover, severe immunosuppression among patients with HIV/TB coinfection correlates with increased optical density in ELISAs (31). This is analogous to the higher Xpert MTB/RIF assay sensitivity observed for patients with smear-positive pulmonary TB than for smear-negative TB patients, presumably due to the increased bacillary burden in sputum (5, 34). It is not clear whether concentration of LAM is higher in morning urine than spot urine and to what extent other host- or pathogen-related factors are associated with increased LAM detection by FujiLAM testing, e.g., presence of nontuberculous mycobacteria, site of TB, and kidney function. Studies of quantitative and urine LAM concentration methods may offer additional insight into this.

Potential limitations of this study include the relatively small number of participants included, which limits the power of the study to detect statistically significant differences, especially in sensitivity and in analysis of subgroup data. We had to exclude several participants from the original cohort because of unavailability of paired spot and early morning urine samples that may represent a selection bias, as those with only one sample available may differ from those with two samples available if too sick or too well to come back to clinic with a second sample. We further allowed a delay of up to 7 days between spot and early morning urine and urine collection at home for outpatients that may have affected both sensitivity and specificity. However, LAM is considered heat and protease stable and does not readily degrade in clinical samples (35), and preclinical studies for FujiLAM did not show cross-reactivity with fast-growing nontuberculous mycobacteria or with microorganisms that could potentially have contaminated urine (15). We sought to minimize the risk of contaminant by careful instructions in sample collection, provision of sterile urine container, and immediate storage of urine in a freezer once received. None of the participants had started TB treatment between sample collection. Our reference standard was limited by sputum Xpert not being available for all patients and use of a low volume of urine (6 mL) for Xpert analysis. A suboptimal reference standard may have misclassified participants, and a number of those with a false-positive FujiLAM result may, in fact, have been true positive, particularly in patients with low CD4 counts. Last, the study findings need to be confirmed

in studies with prospective testing of fresh urine, although Broger et al. did find near-equivalent results for FujiLAM testing of frozen versus fresh urine samples (19).

In conclusion, this study indicates that FujiLAM testing performs equivalently on spot and early morning urine samples. Sensitivity could be increased with a two-sample strategy but at the risk of lower specificity, although the lower specificity may reflect limitations of the reference standard to correctly identify all cases of TB. These data can inform future guidelines and clinical practice around FujiLAM. The minimal difference does not warrant a programmatic challenge of a second early morning sample outside those who are severely sick or immunocompromised. However, this would need to be assessed in larger clinical trials.

## MATERIALS AND METHODS

**Design, study setting, and participants.** The study is based on cryopreserved samples and clinical data collected as part of the DETECT-TB prospective study cohort of participants recruited when presenting for antiretroviral therapy (ART) initiation at Korle-Bu Teaching Hospital in Accra, Ghana, between January 2013 and March 2014 (36). We previously reported the comparative diagnostic accuracy of FujiLAM and AlereLAM assays for TB diagnosis based on a single urine sample (16). Here, we compare diagnostic accuracy of FujiLAM testing for spot versus early morning urine and for a one-sample versus two-sample strategy. The accuracy of test strategies was evaluated against a microbiological and composite reference standard for TB.

The DETECT-TB study enrolled participants consecutively from the outpatient and inpatient department regardless of the presence of TB-related signs and if HIV-positive (seropositive), ≥18 years of age, and referred for ART initiation (i.e., WHO clinical stage 3 or 4, CD4 count ≤ 350 cells/$\mu$L, or pregnant as per recommendations at the time of the study) (37). We excluded participants if they had received anti-TB treatment or preventive TB treatment within the last 2 months. For analyses, we excluded participants who were unable to produce both sputum and urine samples for mycobacterial testing. For the current study, we further excluded those who did not have paired (spot and morning) urine samples available for LAM testing. We followed participants for 6 months to record vital status.

**Data collection and sample procedures.** Upon enrollment, we collected sociodemographic and clinical characteristics. Participants were asked to provide a spontaneously voided urine sample in a sterile container at inclusion in the study, i.e., spot urine. They were provided with a second sterile container to collect and return the first voided early morning urine sample the following day, i.e., morning urine. Participants were instructed to return the morning urine the same morning as collected and keep the sample out of direct sunlight during transport. For outpatients, morning urine was collected at home, and for inpatients, morning urine was collected early morning at the bedside. A pragmatic delay in returning early morning urine up to 7 days from enrollment was accepted to accommodate those outpatients not able to come back with a sample the following morning and to ensure that participants enrolled on a Friday could return morning samples on the next possible weekday. Samples were immediately brought to the Department of Medical Microbiology, University of Ghana, and stored at −20°C and later shipped on dry ice to the Research Institute of Tuberculosis/Japan Anti-Tuberculosis Association in Tokyo for LAM testing.

FujiLAM testing was done in batches on thawed urine samples (spot and early morning) between January and March 2019 (lot ID 98004). Here, approximately 200 $\mu$L of urine was added to the reagent tube up to an indicator line, mixed, and incubated for 40 min at ambient temperature. After mixing again, two drops of sample were added to the test strip followed by sequential pressing of two buttons on the test strip. A result was available within 50 to 60 min by visual inspection of the test lines. The FujiLAM test results were interpreted by two independent readers blinded to each other's observations and the results of the TB reference tests.

For a two-sample strategy (spot and early morning urine), the FujiLAM test was considered positive if at least one of the two urine samples tested positive.

For TB reference standard testing, spontaneously produced sputum samples were collected at random at enrollment and as an early morning sputum. Sputum samples were sent for mycobacterial microscopy, culture using both solid Löwenstein-Jensen medium, and mycobacteria growth indicator tube liquid culture (MGIT; 960 system, BD Diagnostics, Sparks, MD, USA) and Xpert MTB/RIF testing. Presence of *M. tuberculosis* in positive cultures was confirmed by acid-fast staining and an anti-MPB64 antibody assay. In addition, Xpert MTB/RIF assay was also performed on 6 mL biobanked urine in batch after thawing and centrifugation of samples. Those performing and interpreting the reference standard analysis were blinded to the result of the FujiLAM test and clinical data.

Participants were categorized as confirmed TB with the detection of *M. tuberculosis* complex by a positive sputum culture and/or Xpert result on any of the sputum or urine samples collected at baseline. Patients were defined as having probable TB if they started empirical TB treatment or had negative baseline TB diagnostic but a positive laboratory result at 2 months follow-up. See case definitions in Table 3.

**Reference standard and statistical analyses.** In analyses performed, patients with confirmed TB (Table 3) were considered positive by the microbiological reference standard positive, while individuals without positive microbiological studies (i.e., probable or no TB; Table 3) were considered negative by the microbiological reference standard. In analyses using a composite reference standard, participants classified as probable TB were combined with confirmed TB to serve as composite reference standard

**TABLE 3** Tuberculosis diagnostic classification[a]

| Category | Description |
|---|---|
| Confirmed TB | Any culture or any Xpert MTB/RIF (baseline) positive for MTB, ≥1 positive sputum culture (solid, liquid) and confirmed MTB complex at baseline, or ≥1 positive Xpert MTB/RIF (sputum or urine) at baseline |
| Probable TB | Any patient not meeting confirmed TB or no TB classification who is started on TB treatment or has positive laboratory findings on 2 months follow-up, empiric TB treatment started by the healthcare provider, or positive sputum culture, sputum Xpert, and/or sputum smear on follow-up |
| No TB | All microscopy, culture, and Xpert MTB/RIF tests negative for MTB, not started on TB treatment, and has negative follow-up tests; all cultures negative (sputum, including follow-up where available); all Xpert negative (sputum and urine, including follow-up where available); all smear microscopy negative (sputum, including follow-up where available), and treatment not initiated by healthcare providers |

[a]MTB, *Mycobacterium tuberculosis* complex; TB, tuberculosis.

positive, while participants with no TB were considered negative by the composite reference standard. The analysis according to diagnostic classification and reference standards is illustrated in Supplementary Figure S1. We performed sensitivity analyses excluding participants classified as no TB that had died or were lost to follow-up before 2 months, as their TB status is difficult to ascertain (Supplementary Figure S1).

Descriptive statistics were used for baseline demographic and clinical characteristics. We calculated point estimates for sensitivity and specificity with 95% confidence intervals (CIs) for different testing strategies based on sample type (spot versus morning urine) and number (one versus two urine samples). Accuracy between test strategies was compared using McNemar's test for paired proportion, and the agreement was further determined with Cohen's kappa coefficient. We predefined stratified analyses by CD4 cells/$\mu$L categories (CD4 $\leq$ 200, CD4 $>$ 200, CD4 $\leq$ 100, CD4 $>$ 100, and CD4 counts of 101 to 200), by patient status (outpatient, inpatient), and by mortality before 6 months of follow-up, as these parameters are known to be correlated with LAM detection (11, 16–18). We further did explorative analyses by sex and for adults with advanced HIV defined by presence of a CD4 cell count of $<$200 cells/$\mu$L or a WHO clinical stage 3 or 4 event (38). Due to the secondary nature of our study, we did no formal sample size calculation.

All analyses were conducted using STATA version 16.1 software.

**Ethics.** All patients enrolled provided written informed consent for participation, and the study was approved by the Institutional Review Board of University of Ghana Medical School (MS-Et./M.4-P 3.3/2012-13) and the Danish National Committee on Health Research Ethics (no. 1302133/1206169).

## SUPPLEMENTAL MATERIAL

Supplemental material is available online only.

**SUPPLEMENTAL FILE 1**, DOCX file, 0.2 MB.

## ACKNOWLEDGMENTS

We thank the participants in the DETECT HIV-TB study. We also thank Kwaku Appiah-Korang Labi for organizing samples before shipment to Japan, as well as the staff at the Department of Medical Microbiology, School of Biomedical and Allied Health, University of Ghana, and at the designated laboratory at Research Institute of Tuberculosis/Japan Anti-Tuberculosis Association for organizing transport and testing of samples. Special thanks to Claudia M. Denkinger at the Division of Tropical Medicine, Center of Infectious Diseases, University Hospital Heidelberg, Heidelberg, Germany, who took part in conceptualization of the study.

The study was supported by FIND and funded by the Global Health Innovative Technology Fund (GHIT grant number G2017-207) and the KfW (grant number 2020 60 457), whereas the DETECT HIV-TB study was funded by a grant from Odense University Hospital Free Research Fund (grant number 11/28764), the Danish AIDS Foundation (grant number F12-10), and the Aase and Ejnar Danielsens Fond (grant number 10-001013).

The funders of the study had no role in study design, data collection, data analysis, data interpretation, or the decision to submit the work for publication.

FIND is a not-for-profit NGO that collaborates in partnerships to develop, evaluate, and implement new diagnostics for LMIC. FIND has product evaluation agreements with FujiFilm and several other private sector companies that design diagnostics and related products for treatment of tuberculosis and other diseases. These agreements strictly define FIND's independence and neutrality vis-à-vis the companies whose products get evaluated and describe roles and responsibilities.

S.B., R.S., S.M., T.B., and I.S.J. had the idea for and designed the study. S.B., J.A., T.B., M.L., J.O., and S.M. analyzed the samples and collected the data. S.B. and J.A. did the analysis and generated the estimates. S.B. drafted the initial version of the manuscript with input from J.A., R.S., J.O., M.L., M.S., S.M., T.B., M.R., and I.S.J. All authors contributed to the revision and correction on multiple iterations of the manuscript. The corresponding author had full access to all data in the study and had final responsibility for the decision to submit for publication.

S.B., I.S.J., J.A., J.O., and M.L. have no conflict of interest to report.

T.B., M.R., and R.S. were affiliated with FIND during the conducting of the study. T.B. reports patents in the field of TB detection and is a shareholder of Avelo Inc. S.M. reports collaboration with FujiFilm for the development of SILVAMP TB LAM and that the study budget received approximately $20,000 in the last 2 years for conduct of FujiLAM tests at RIT/JATA laboratory under the contract with FIND.

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
