## [Reviewer comments · Microbiology Spectrum]

Microbiology Spectrum

Diagnostic accuracy of urine lipoarabinomannan testing in early morning urine versus spot urine for diagnosis of tuberculosis among people with HIV

Stephanie Bjerrum, Johanna Aahsberg, Rita Szekely, Japheth Opintan, Margaret Lartey, Maunank Shah, Satoshi Mitarai, Tobias Broger, Morten Ruhwald, and Isik Johansen

Corresponding Author(s): Stephanie Bjerrum, University of Southern Denmark

Review Timeline:

Submission Date:	January 29, 2022
Editorial Decision:	February 18, 2022
Revision Received:	February 24, 2022
Accepted:	March 4, 2022

Editor: Tomefa Asempa

Reviewer(s): The reviewers have opted to remain anonymous.

Transaction Report:

DOI: <https://doi.org/10.1128/spectrum.00208-22>

February 18, 2022

Dr. Stephanie Bjerrum
University of Southern Denmark
Department of Clinical Research, Infectious Diseases
Winsløwparken 19
Odense 5000
Denmark

Re: Spectrum00208-22 (Diagnostic accuracy of urine lipoarabinomannan testing in early morning urine versus spot urine for diagnosis of tuberculosis among people with HIV)

Dear Dr. Stephanie Bjerrum:

Thank you for submitting your manuscript to Microbiology Spectrum. As you will see your paper is very close to acceptance. Please modify the manuscript along the lines I have recommended. As these revisions are quite minor, I expect that you should be able to turn in the revised paper in less than 30 days, if not sooner. If your manuscript was reviewed, you will find the reviewers' comments below.

When submitting the revised version of your paper, please provide (1) point-by-point responses to the issues I raised in your cover letter, and (2) a PDF file that indicates the changes from the original submission (by highlighting or underlining the changes) as file type "Marked Up Manuscript - For Review Only". Please use this link to submit your revised manuscript. Detailed instructions on submitting your revised paper are below.

Link Not Available

Sincerely,

Tomefa Asempa

Editor comments:

Appreciate response to reviewers however I find explanation of the impact of gender on discordant results to be subpar and more insights can be provided.

You note further studies may provide insights but there is robust PREVIOUS data. What is the difference between current study (population etc) and the numerous (Bjerrum et al. 2020, Broger et al. 2020) studies that have come before it and not noted this trend for both Alere and Fuji.

Preparing Revision Guidelines

- point-by-point responses to the issues I raised in your cover letter
- Upload a compare copy of the manuscript (without figures) as a "Marked-Up Manuscript" file.
- Each figure must be uploaded as a separate file, and any multipanel figures must be assembled into one file.
- Manuscript: A .DOC version of the revised manuscript
- Figures: Editable, high-resolution, individual figure files are required at revision, TIFF or EPS files are preferred

Please return the manuscript within 60 days; if you cannot complete the modification within this time period, please contact me. If you do not wish to modify the manuscript and prefer to submit it to another journal, please notify me of your decision immediately so that the manuscript may be formally withdrawn from consideration by Microbiology Spectrum.

February 24th, 2022

Re: Spectrum00208-22

*Dear Mr Tomefa,
Editor, Microbiology Spectrum*

Thank you for reviewing our manuscript and response to reviewers. Below, we address your comments point-by-point.

Editor comments: Appreciate response to reviewers however I find explanation of the impact of gender on discordant results to be subpar and more insights can be provided.

Response: Thank you very much for reviewing our manuscript and our earlier response to reviewers. The finding of gender being associated with a discordant FujiLAM result was a surprise to our team and not a result we have seen reported before. We know of only one other study that assessed the effect of early morning urine of accuracy of AlereLAM (Gina et al. BMC inf dis 2017 = Reference 29 in the manuscript) but have not seen this analysis done for FujiLAM. The study by Gina et al. found an increase in AlereLAM sensitivity using early morning urine, but did not include gender as a variable in their analyses of predictors for increased AlereLAM sensitivity and also did not assess specificity. In response to earlier reviewers' comments, we included in the Appendix Table A2 additional variables to assess if there were obvious host characteristics like proteinuria, higher creatinine, pregnancy or differences in TB status by reporting Urine Xpert and Alere LAM results among participants that could explain the discordant results. We clarified this in the revised manuscript.

Editor comments: You note further studies may provide insights but there is robust PREVIOUS data. What is the difference between current study (population etc) and the numerous (Bjerrum et al. 2020, Broger et al. 2020) studies that have come before it and not noted this trend for both Alere and Fuji.

Response: We agree that there is robust data on accuracy in particular for AlereLAM and equally evolving data for FujiLAM. However, previous LAM accuracy studies (AlereLAM and FujiLAM) have not reported accuracy data by gender, and focus of subpopulation analysis have mainly been on HIV status, CD4 cell count, admission status and other factors related to severeness or site of HIV/TB disease. Some early studies assessed but did not find gender to be associated with AlereLAM positivity (based on spot urine samples) eg Nakiyingi et al J Acquir Immune Defic Syndr 2014; Lawn et al. BMC Medicine 2017; wood et al BMC Infectious Diseases 2012. The study population in the current manuscript is equal to the study population in Bjerrum et al 2020 and represents one of the five cohorts in Broger et al. 2020, but these previous studies focused on comparison of AlereLAM and FujiLAM accuracy and did not include data on the effect of early morning urine or two-sample strategy as is the focus of the current manuscript. We revised the manuscript to highlight that we could make use of existing dataset to assess the effect of gender on urine LAM and should be attentive to include gender as a possible significant variable in future LAM studies.

Thank you again for considering our revised manuscript for publication in Microbiology Spectrum. We look forward to hearing from you.

Yours sincerely,

Dr Stephanie Bjerrum

March 4, 2022

Dr. Stephanie Bjerrum
University of Southern Denmark
Department of Clinical Research, Infectious Diseases
Winsløwparken 19
Odense 5000
Denmark

Re: Spectrum00208-22R1 (Diagnostic accuracy of urine lipoarabinomannan testing in early morning urine versus spot urine for diagnosis of tuberculosis among people with HIV)

Dear Dr. Stephanie Bjerrum:

Appreciate response and revisions.

Your manuscript has been accepted, and I am forwarding it to the ASM Journals Department for publication. You will be notified when your proofs are ready to be viewed.

Sincerely,

Tomefa Asempa
Editor, Microbiology Spectrum
